# Prediction of Subsequent Contralateral Patellar Dislocation after First-Time Dislocation Based on Patellofemoral Morphologies

**DOI:** 10.3390/jcm12010180

**Published:** 2022-12-26

**Authors:** Jiaxing Chen, Qiaochu Li, Sizhu Liu, Lin Fan, Baoshan Yin, Xinyu Yang, Linbang Wang, Zijie Xu, Jian Zhang, Zhengxue Quan, Aiguo Zhou

**Affiliations:** 1Department of Orthopedics, The First Affiliated Hospital of Chongqing Medical University, Chongqing 400016, China; 2Orthopedic Laboratory, Chongqing Medical University, Chongqing 400016, China; 3Department of Radiology, The First Affiliated Hospital of Chongqing Medical University, Chongqing 400016, China; 4Department of Orthopedics, Peking University Third Hospital, Beijing 100191, China; 5Sports Medicine Department, Beijing Key Laboratory of Sports Injuries, Peking University Third Hospital, Beijing 100191, China

**Keywords:** patellar dislocation, contralateral patellofemoral joint, anatomic risk factor, TT-RA distance, HKA angle

## Abstract

The subsequent dislocation of a contralateral patellofemoral joint sometimes occurs after a first-time lateral patellar dislocation (LPD). However, the anatomic risk factors for subsequent contralateral LPD remain elusive. This study included 17 patients with contralateral LPD and 34 unilateral patellar dislocators. The anatomic parameters of the contralateral patellofemoral joints were measured using CT images and radiographs that were obtained at the time of the first dislocation. The Wilcoxon rank-sum test was performed, and a binary regression model was established to identify the risk factors. The receiver operating characteristic curves and the area under the curve (AUC) were analyzed. The tibial tubercle-Roman arch (TT-RA) distance was significantly different between patients with and without contralateral LPD (24.1 vs. 19.5 mm, *p* < 0.001). The hip–knee–ankle (HKA) angle, patellar tilt, congruence angle, and patellar displacement were greater in the study group than in the control group (*p* < 0.05). The TT-RA distance revealed an OR of 1.35 (95% CI (1.26–1.44]), *p* < 0.001) and an AUC of 0.727 for predicting contralateral LPD. The HKA angle revealed an OR of 1.74 (95% CI (1.51–2.00), *p* < 0.001) and an AUC of 0.797. The Patellar tilt, congruence angle, and patellar displacement had AUC values of 0.703, 0.725, and 0.817 for predicting contralateral LPD, respectively. In conclusion, the contralateral patellofemoral anatomic parameters were significantly different between patients with and without subsequent contralateral LPD. Increased TT-RA distance and excessive valgus deformity were risk factors and could serve as predictors for contralateral LPD. At first-time dislocation, the abnormal position of the patella relative to the trochlea may also be an important cause of subsequent LPD.

## 1. Introduction

Lateral patellar dislocation (LPD) is a common disorder that mainly affects the unilateral patellofemoral joints of female adolescents [1,2]. Overall, 5.4–5.8% of patients with unilateral LPD will suffer the dislocation of contralateral patellofemoral joints at some point during their life [3,4]. Although low incidence of contralateral LPD is observed, it has a significant influence, in terms of psychological and physical trauma, on patients who suffer subsequent contralateral LPD. Preventing subsequent contralateral LPD is crucial, but the etiology of its initiation remains elusive.

In recent years, risk factors for LPD or its recurrence have been well identified, and various anatomic abnormalities with regard to patellofemoral joints have been found to be responsible for LPD [5]. Trochlear dysplasia was reported as the most significant risk factor in the production of patellar instability as it decreases patellotrochlear congruence, with a morbidity of more than 90% in patients with LPD [6]. In addition, some specific deformities may occur concurrently, such as the excessive lateralization of the tibial tubercle, femoral or tibial rotational malformation, coronal malalignment, patella alta, and patella tilt, which could aggravate patellar instability by causing imbalanced forces around patellofemoral joints [7,8,9].

To the best of our knowledge, anatomic risk factors for subsequent contralateral LPD remain unclear; moreover, even research into contralateral-patellofemoral-joint-related morphologies is scarce [10,11]. Patients with skeletal immaturity at first-time dislocation have the highest risk of contralateral LPD [4]. The occurrence of ipsilateral LPD was reported an important risk factor for contralateral dislocation [12]. Trochlear dysplasia of dislocated knees can sometimes predict contralateral LPD after first-time dislocation [4]. Dejour et al. [13] found that dislocators potentially had a trochlear dysplasia in the contralateral unaffected knees. In addition, Simonaitytė et al. [10] reported a 24.1% incidence of trochlear dysplasia in contralateral knees. Our previous study revealed that patella alta and trochlear dysplasia could be traced in contralateral un-dislocated knees [11]. Demehri et al. [14] uncovered anatomic abnormalities in contralateral asymptomatic patellofemoral joints in patients with unilateral LPD.

The hypothesis of this study was that patients who suffered subsequent contralateral LPD after a first-time dislocation were characterized by more severe contralateral-patellofemoral-joint-related skeletal deformities than patients without contralateral LPD. Given the insufficient knowledge in the literature, the purpose of this study was twofold: the first was to verify the difference in the anatomic parameters of unaffected knees at a first-time dislocation between patients with and without contralateral LPD; the second was to identify any anatomic variations that could contribute to or predict subsequent contralateral LPD.

## 2. Materials and Methods

### 2.1. Study Population

Approval from the Institutional Review Board (IRB) of our hospital was obtained, and the informed consent was waived by the IRB. (IRB NO. 2022-K26). A total of 185 consecutive patients admitted to our institution with unilateral LPD and asymptomatic contralateral patellofemoral joints from January 2015 to December 2020 were identified. The inclusion criteria for this retrospective case-control study were as follows: patients who suffered contralateral LPD by November 2022. In total, 19 patients were considered eligible for inclusion in this study. The exclusion criteria were as follows: patients with ipsilateral traumatic or habitual patellar dislocation; patients who did not have a CT scan of the contralateral hip, knee, and ankle joints simultaneously at the time of their first dislocation; patients without weight-bearing full-leg anteroposterior radiographs or lateral knee radiographs; patients with a history of contralateral bone fracture or surgery that may influence the measurements; patients with severe epiphysitis of the femur.

Among the nineteen patients, two subjects who lacked the necessary CT images and radiographs were excluded. As a result, 17 patients with subsequent contralateral LPD were designated as the study group. We conducted 1:2 matching for the patients without contralateral LPD during the same period following the above exclusion criteria. Then, 34 patients matched by age (age at first-time dislocation) and sex, who did not have any contralateral-patellofemoral-joint-related symptoms, were included in the control group. The subjects’ inclusion flowchart is shown in Appendix A. By the time we submitted this article, the mean follow-up time of the 185 consecutive patients with unilateral LPD was 45 ± 20 months. All of the patients were regularly followed up.

### 2.2. Computed Tomography Technique

CT Images were obtained using a scanner (Somatom Sensation, Siemens Healthcare, Forchheim, Germany) in our hospital within seven days after the first dislocation, ranging from the bilateral ilium to the toes. Patients were in the supine position with the bilateral lower limbs in full extension and the foot positioned in 90° flexion. The scanning parameters were as follows: tube voltage, 130 kVp; tube current 110–140 mAs; scanning layer thickness and layer spacing, 1 mm; matrix, 512 × 512 pixels. The field of view varied with the individual characteristics of the patients, ranging from 220 to 450 mm.

### 2.3. Radiological Assessment

A total of 51 patients (17 in the study group and 34 in the control group) were available for radiological assessment. CT images of the contralateral lower limbs, weight-bearing full-leg anteroposterior radiographs, or lateral knee radiographs at the first instance of dislocation were retrospectively collected. Trochlear dysplasia, tibial tubercle lateralization, segmental femoral anteversion, knee joint rotation, tibial torsion, patellar tilt, patellar height, coronal malalignment, congruence angle, lateral patellar displacement, and posterior condylar angle (PCA) were measured by an experienced orthopedist and a well-trained radiologist in a blinded and randomized fashion using the picture archiving and communication system (PACS). In the event of any major disputes about the measuring results, especially the Dejour classification of trochlear dysplasia, a discussion with another experienced orthopedist was conducted until a consensus was reached. All of the measurements were conducted two weeks later to assess intra-observer reliability.

#### 2.3.1. Trochlear Dysplasia

Dejour classification, a commonly used four-grading system for evaluating trochlear dysplasia, consists of Type A: shallow trochlea; Type B: flat or convex trochlea; Type C: a convex medial trochlear wall; and Type D: cliff patterns [15]. Lateral trochlea inclination (LTI) and trochlear depth were reported as reliable and objective parameters for quantifying trochlear dysplasia [16]. LTI was measured by the angle between a line tangent to the lateral aspect of the trochlea (LTF) and the surgical transepicondylar axis (SEA) (Figure 1A). The trochlear depth was the depth formed between the medial and lateral femoral trochlear facets (Figure 1B).

#### 2.3.2. Tibial Tubercle Lateralization

Tibial tubercle lateralization was calculated by the tibial tubercle-Roman arch (TT-RA) distance that we proposed previously, which was demonstrated to be more reliable than the tibial tubercle-trochlear groove (TT-TG) distance [17]. Briefly, the highest portion of the Roman arch was identified on the axial CT slice showing intact femoral condyles, and the middle portion of the bony tibial tubercle at the insertion of the patellar tendon was regarded as the landmark for measuring the TT-RA distance (Figure 2).

#### 2.3.3. Femoral Anteversion

Femoral malrotation was assessed using a recently proposed method [18]. Briefly, segmental femoral torsion parameters (total, neck, mid, and distal torsion) were measured by four independent lines: the proximal femoral head-neck-axis, the femur-lesser trochanter line, the tangent of the distal/posterior femur, and the SEA (Figure 3). The total femoral anteversion was defined as the angle between the proximal femoral head-neck axis and SEA, with a value of more than 20.4° indicating a pathology.

#### 2.3.4. Knee Joint Rotation and Tibial Torsion

Knee joint rotation is the angle between the posterior femoral condylar reference line (PCRL) and the dorsal tibia condylar line (Appendix A) [19]. Tibial torsion was assessed by measuring the angle between the dorsal tibia condylar line and a line through the medial and lateral malleolus [20]. PCA was assessed by the angle formed between the SEA and the PCRL, and the length of the SEA was defined as the transepicondylar width (TEW) [18].

#### 2.3.5. Patellar Height and HKA Angle

The Insall–Salvati index was calculated to assess patellar height and was defined as the ratio between the length of the articular surface of the patella and the length of the patellar tendon on the lateral plain radiograph, with a value of >1.2 indicating patella alta (Appendix A) [7]. Coronal malformation was evaluated by the hip–knee–ankle angle: the angle between the femoral and tibial mechanical axis on weight-bearing full-leg anteroposterior radiographs [21], and a value of >0 degree was referred to valgus deformity in this study.

#### 2.3.6. Patella Position Relative to the Trochlea

Patella tilt was assumed to be the angle between the PCRL and patella width line (Figure 4A) [22]. The congruence angle was the angle between the line bisecting the sulcus angle and the line connecting the lowest portion of the sulcus to the apex of the patella ridge (Figure 4B) [23]. Patellar displacement measured the positive values indicating lateral translation and was defined as the distance between the patellar medial edge and the medial femoral condyle (Figure 4C) [24].

### 2.4. Statistical Analysis

The average value of each parameter measured by both observers was used for the final statistical analysis, which was conducted independently via SPSS software (Version 21.0; IBM Corp, Armonk, NY, USA) by a well-trained orthopedist. The TT-RA distance was normalized by TEW to reduce individual differences. The inter- and intra-observer correlation coefficients (ICCs) and weighted kappa analysis were conducted, with a value of >0.75 indicating excellent agreement. Because of the small sample size of this study, all continuous data were presented as the median and interquartile range (IQR). The Chi-square test and Wilcoxon rank-sum test were performed to identify the differences in anatomic parameters between the two groups. The binary logistic regression model was established to identify the anatomic risk factors for contralateral LPD. Receiver-operating characteristic curves (ROC) and the area under the curve (AUC) were used to evaluate the diagnostic ability of each parameter for subsequent contralateral LPD after a first-time dislocation. The Youden index of any parameter with an AUC of more than 0.7 was calculated to identify its sensitivity and specificity.

Post hoc analysis was performed using G-Power software (version 3.1.9.4, Heinrich-Heine-Universitat Dusseldorf, Dusseldorf, Germany). For a large effect size of 1.01 according to the TT-RA distance in the two groups, a power of 0.95 was calculated (n_1_ = 17, n_2_ = 34; alpha, 0.05).

## 3. Results

In total, 10.3% (19/185) of patients suffered contralateral LPD after first-time dislocation, with a mean follow-up time of 45 months; 17 patients with subsequent contralateral LPD and 34 patients without contralateral LPD or any patellofemoral symptoms were included in this study. The demographic data of the patients in the two groups are shown in Table 1. The interval time between the first-time dislocation and the time of contralateral dislocation varied, ranging from 13 months to 75 months. The follow-up time for the control group was 41 (IQR, 14) months. Overall, 35.3% of patients in the study group had skeletal immaturity at their first-time dislocation. The ICCs and 95% confidence interval (CI) of each measurement are shown in Table 2. All the measurements showed good to excellent inter-and intra-observer agreements (ICCs > 0.75).

The differences in the anatomic parameters between the two groups are shown in Table 3. Severe trochlear dysplasia (Type B–D) represented a larger proportion of the study group than the control group (94.1% vs. 88.2%, *p* < 0.001). LTI and trochlear depth were smaller in the study group (11.9° and 3.8 mm, respectively) than in the control group (13.9° and 4.1 mm, respectively) (*p* < 0.001). The TT-RA distance and the ratio of TT-RA/TEW were greater in patients with contralateral LPD (24.1 mm and 32.7%, respectively) than in patients without contralateral LPD (19.5 mm and 27.6%, respectively). The median value of the HKA angle in the study group was 1.9°, compared to 0.8° in the control group (*p* < 0.001). Patellar tilt, congruence angle, and patellar displacement were greater in the study group than in the control group (*p* < 0.001).

The results of the unadjusted regression model (simple analysis using continuous data) are shown in Figure 5. LTI and trochlear depth revealed significant ORs of 1.21 and of 1.42 with regard to subsequent contralateral LPD, respectively (*p* < 0.001). The TT-RA distance and HKA angle were associated with a contralateral LPD with an OR of 1.35 (95% CI (1.26–1.44) *p* < 0.001) and an OR of 1.74 (95% CI (1.51–2.00), *p* < 0.001), respectively. Patellar tilt, the congruence angle, and patellar displacement showed significant correlations with subsequent contralateral LPD, with OR values of 1.13, 1.57, and 2.74, respectively (*p* < 0.001). The distal femoral torsion and tibial rotation also revealed significant ORs of 1.06 (*p* = 0.011) and of 1.05 (*p* = 0.002), respectively.

ROC curves were analyzed to calculate the diagnostic capacity of these parameters for subsequent contralateral LPD (Table 4 and Figure 6). LTI, trochlear depth, distal femoral torsion, and tibial rotation had AUCs of <0.7 (*p* > 0.05). The TT-RA/TEW had an AUC of 0.741 for contralateral LPD, with a cutoff value of 29.5% (82.3% sensitivity and 61.8% specificity, *p* = 0.006); the same was true for the results of the TT-RA distance. Patellar displacement and the HKA angle revealed significant AUCs of 0.817 and of 0.797, with cutoff values of 9.2 mm (sensitivity 88.2% and specificity 64.7%) and of 1.3° (sensitivity 82.4% and specificity 70.6%), respectively (*p* < 0.001). Patellar tilt and the congruence angle had an AUC of 0.703 and of 0.725 for predicting contralateral LPD, respectively.

## 4. Discussion

The most important findings of this study are as follows. The contralateral patellofemoral anatomic parameters were significantly different between patients with and without subsequent contralateral LPD; these parameters include trochlear dysplasia, valgus malalignment, and tibial tubercle lateralization. The TT-RA distance and HKA angle are verified as risk factors and can serve as potential predictors for subsequent contralateral LPD. In addition, although contralateral patellofemoral joints are asymptomatic at first-time dislocation, the abnormal positioning of the contralateral patella does exist, and has characteristics such as excessive patellar tilt, the congruence angle, and patellar displacement, which are implicated in subsequent contralateral LPD.

Skeletal abnormalities were demonstrated to be implicated in LPD. Trochlear dysplasia, excessive lateralization of the tibial tubercle, femoral anteversion, and coronal malalignment were reported as the main risk factors for LPD in that they reduce patellotrochlear congruence and increase the lateral vector force of the patella [15,25]. Previous literature has reported that 5.4% to 5.8% of patients with unilateral LPD could suffer subsequent contralateral LPD after first-time dislocation [3,4]. In this study, the incidence of contralateral LPD was 10.3%. The severe anatomic abnormalities with regard to contralateral patellofemoral joints may be a reason for such higher incidence. It yields a great significance to explore the role of contralateral anatomic abnormalities in subsequent LPD for both disease recognition and prevention.

To the best of our knowledge, the literature regarding the morphologies of contralateral patellofemoral joints in patients with unilateral LPD is scarce. Anatomic abnormalities of contralateral asymptomatic patellofemoral joints were identified in patients with unilateral LPD [14]. The contralateral asymptomatic knees had increased patellar heights and excessive lateralization of the tibial tubercle in patients with unilateral patellar dislocation [11]. Dejour et al. [13] explained that a large proportion of patients with trochlear dysplasia in one knee potentially had a trochlear dysplasia in the contralateral intact knee. In addition, Simonaitytė et al. [10] reported a 24.1% and 84.5% incidence of trochlear dysplasia and patellar alta in contralateral knees, respectively.

Previous literature focused on risk factors affecting ipsilateral LPD recurrence after first-time dislocation [26], but the risk factors for subsequent contralateral LPD have not been adequately studied, especially anatomic ones. Patients with skeletal immaturity at the first-time dislocation have the highest risk of contralateral LPD [4]. Parikh et al. [27] reported that the presence of trochlear dysplasia in the affected knees had an OR of 8.7 for subsequent contralateral patellar instability. Christensen et al. [4] considered trochlear dysplasia of the ipsilateral dislocated knees as a risk factor for contralateral LPD after first-time dislocation. It remains unclear why trochlear dysplasia in ipsilateral knees could predict contralateral LPD; the significant correlations between bilateral skeletal features may be one reason [11]. In this study, trochlear dysplasia of contralateral knees was found to be more severe in patients who suffered subsequent contralateral LPD than in patients without contralateral LPD, which was considered as a risk factor but was of limited predictive value for subsequent contralateral LPD (AUC < 0.7).

The TT-RA distance and HKA angle could reliably reflect tibial tubercle lateralization and coronal malalignment, respectively, which could contribute to patellar instability by increasing patella lateralization and the pressure of the lateral trochlear facet [28,29]. Christensen et al. [4] reported that the TT-TG distance in the affected knees could not predict subsequent contralateral LPD. In addition, the TT-TG distance in the affected knees was not correlated with contralateral anatomic abnormalities in patients with unilateral patellar dislocation [11]. In this study, excessive TT-RA distance in the contralateral knees at first-time dislocation was verified as a risk factor for subsequent contralateral LPD, with a fair predictive ability. When the value of the TT-RA distance was 20.0 mm, the sensitivity of subsequent contralateral LPD would be 82.3%, and for every 1 mm increase in the TT-RA distance, the risk would increase 1.4 times. To the best of our knowledge, this study is the first to report the HKA angle of contralateral knees in patients with subsequent LPD. When the value of the HKA angle was 1.3°, it revealed a high sensitivity (82.4%) for predicting subsequent contralateral LPD; in addition, for every 1 degree increase in the HKA angle, the risk of subsequent contralateral LPD was 1.7-fold higher.

Rotational malformation of lower extremities, such as excessive femoral anteversion and tibial external rotation were likely to be identified in patients with LPD [30] and served as risk factors for patellar instability [31]. We have demonstrated that the differences in femoral anteversion and tibial rotation between ipsilateral and contralateral knees are not significant in patients with unilateral LPD [11]. In this study, except for tibial rotation, differences in the rotational parameters between patients with and without subsequent contralateral LPD were not significant, indicating that lower limb malrotation may exist independently of contralateral LPD. On the other hand, tibial external rotation could not serve as a risk factor or predictor for subsequent contralateral LPD because of a relatively small AUC and OR value, which warrants further investigation to include a large proportion of patients.

The abnormal position of the patella relative to the femoral trochlea, such as excessive patellar tilt, congruence angle, and patellar displacement, were reported to be implicated in LPD by reducing patellotrochlear congruence and aggravating patellar instability [22,23,24]. These parameters were significantly different between patients with and without contralateral LPD and were verified as predictive factors for contralateral LPD, with an OR of 1.6 to 2.7. As it stands, even though contralateral patellofemoral joints were asymptomatic, the abnormal position of contralateral patella did exist at the time of first dislocation, which may contribute to contralateral LPD. With regard to patellar height, Parikh et al. [27] considered ipsilateral patella alta as a risk factor for contralateral LPD. Based on our previous research, the correlations between the patellar height in ipsilateral affected knees and the anatomic abnormalities in contralateral asymptomatic knees were not significant [11]. Moreover, Simonaitytė et al. [10] demonstrated that the mean value of the Blackburne–Peel index was 1.2 in contralateral intact knees, compared to 1.3 in dislocated knees (*p* > 0.05). In this study, contralateral patellar height measured using the Insall–Salvati index was not significantly different between patients with and without contralateral LPD (1.32 vs. 1.34), indicating that it is of limited value for predicting subsequent contralateral LPD. These results should be verified via a study to include a larger sample size.

Our study had some limitations. First, this study only included anatomic parameters; the roles of clinical data in contralateral LPD, such as injury mechanism, exercise intensity, and the relaxation of multiple ligaments were not discussed. Second, the sample size in the study group was relatively small due to the low incidence of subsequent contralateral LPD, meaning that a study including a large population is necessary. Third, patients in the control group may suffer contralateral LPD in the future, which could result in a potential bias in the results. Further follow-up is necessary. Fourth, patients with skeletal maturity and immaturity were not completely separated, and the roles of age and gender at first-time dislocation in subsequent contralateral LPD were not reported in this study. Fifth, the anatomic parameters of the ipsilateral patellofemoral joints at the first-time dislocation in patients with and without contralateral LPD were not compared. Sixth, the question of whether the abnormal anatomic parameters could influence the time of suffering subsequent contralateral LPD warrants further investigation.

## 5. Conclusions

Contralateral patellofemoral anatomic parameters were significantly different between patients with and without subsequent contralateral LPD. Increased TT-RA distance and excessive valgus deformity were risk factors and could serve as predictors for contralateral LPD. At first-time dislocation, the abnormal position of the patella relative to the trochlea may also be an important cause of subsequent LPD.

## Figures and Tables

**Figure 1 jcm-12-00180-f001:**
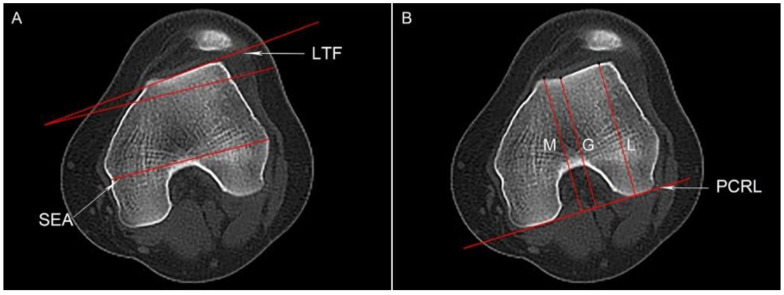
Measurement of the lateral trochlea inclination (LTI) and trochlear depth. (**A**) The line through the sulcus of the medial epicondyle and the prominence of the lateral epicondyle (SEA) and a line subtended from the surface of the lateral trochlea (LTF) are shown. The red dotted line is parallel to the SEA, and LTI is the angle between the SEA and LTF. (**B**) The highest points of the lateral/medial trochlear facets and the deepest point of the trochlea were identified, showing their vertical distance to the posterior condylar reference line (PCRL), L, M, and G, respectively. Trochlea depth is calculated according to the following formula: [(L + M)/2 − G].

**Figure 2 jcm-12-00180-f002:**
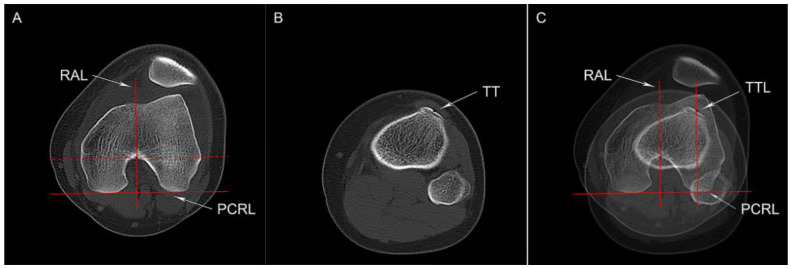
Tibial tubercle-Roman arch (TT-RA) distance (**A**) The posterior condylar reference line (PCRL), a line parallel to the PCRL and tangent to the Roman arch (the red dotted line), and a line that passes through the tangent point (the black dot) and perpendicular to the PCRL (RAL) are shown. (**B**) Axial CT image with the completed contact of the tibial tendon, showing the center of tibial tuberosity (TT). (**C**) The line perpendicular to the PCRL and parallel to RAL is drawn through the TT (TTL), and the distance between the RAL and TTL is defined as the TT-RA distance.

**Figure 3 jcm-12-00180-f003:**
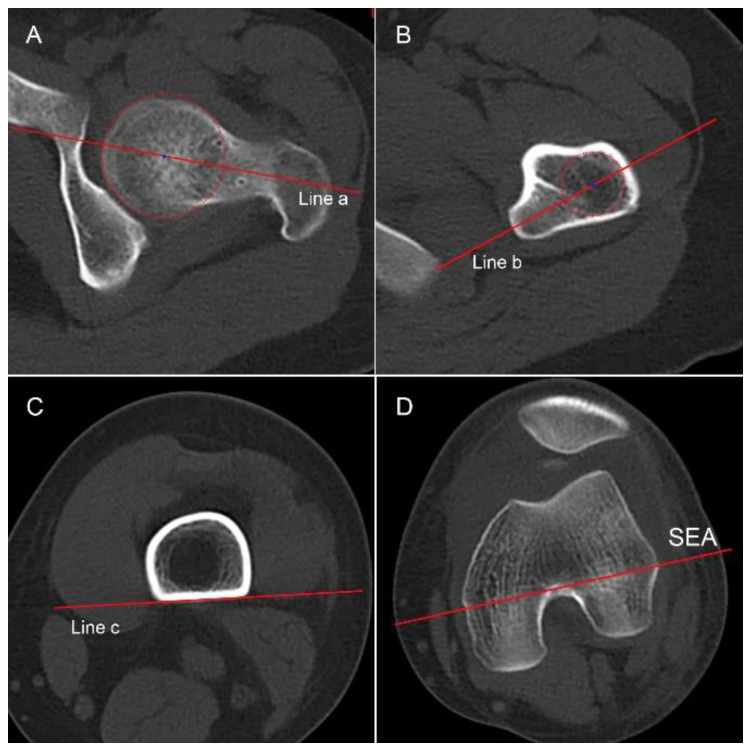
Four segments of femur are selected (**A**) The femoral head–neck axis is drawn through the center of the femoral head and neck (Line a). (**B**) The femur–lesser–trochanter line is drawn through the center of the femur and the midpoint of the lesser trochanter (Line b). (**C**) Line c is tangent to the posterior aspect of the femur on the slice just above the attachment of the gastrocnemius. (**D**) A line through the sulcus of the medial epicondyle and the prominence of the lateral epicondyle (SEA) is shown. The angles formed between Line a and Line b, Line b and Line c, Line c and SEA, and Line a and SEA are regarded as the neck torsion, mid torsion, distal torsion, and total femoral torsion, respectively.

**Figure 4 jcm-12-00180-f004:**
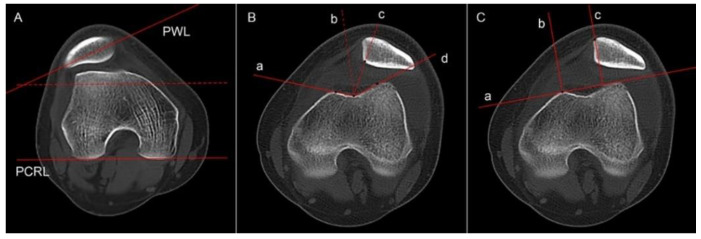
(**A**) Patella tilt is the angle formed between a line through the patella width (PWL) and a line tangent to the bilateral posterior condyles (PCRL). The red dotted line is parallel to the PCRL. (**B**) Congruence angle is defined as the angle between the sulcus angle bisector (dotted line b) and the line connecting the deepest portion of the sulcus and the apex of the patella ridge (dotted line c). (**C**) Patellar displacement is the distance between two vertical lines of the line tangent to the bilateral anterior condyles (line a), the one passing through the highest portion of the medial facet (line b), and the other one passing through the medial edge of the patella (line c).

**Figure 5 jcm-12-00180-f005:**
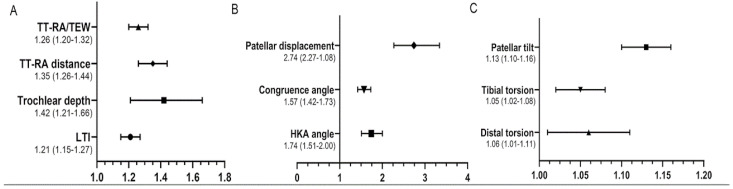
The results of the unadjusted binary regression model, showing the Odd Ratio and 95% confidence interval. (**A**) The dendrogram of TT-RA/TEW, TT-RA distance, trochlear depth, and LTI. (**B**) The dendrogram of patellar displacement, congruence angle, and HKA. (**C**) The dendrogram of patellar tilt, tibial torsion, and distal torsion. TT-RA, tibial tubercle-Roman arch distance; TEW, transepicondylar width; LTI, lateral trochlea inclination; HKA, hip–knee–ankle angle.

**Figure 6 jcm-12-00180-f006:**
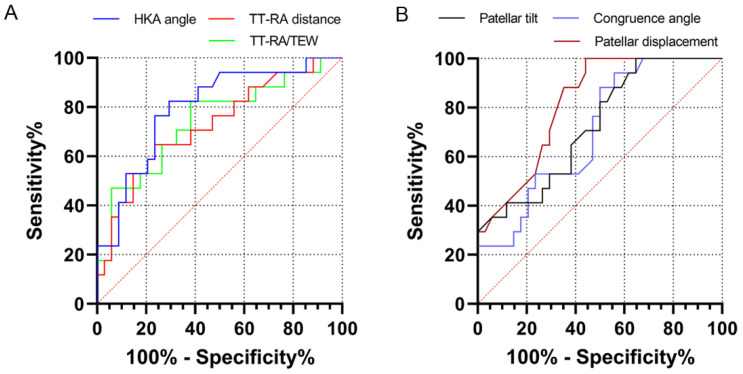
Receiver operating characteristic (ROC) curves of the anatomic parameters. (**A**) The ROC curves of the TT-RA distance, HKA angle, and TT-RA/TEW. (**B**) The ROC curves of patellar tilt, congruence angle, and patellar displacement. TT-RA, tibial tubercle-Roman arch distance; HKA, hip–knee–ankle angle; TEW, transepicondylar width.

**Table 1 jcm-12-00180-t001:** Demographic characteristics of the participants.

	Study Group (*n* = 17)	Control Group (*n* = 34)	*p* Value
Sex, *n*			n.s
Female	12	24	
Male	5	10	
Age, median (IQR), y	19.2 (7.0)	19.0 (7.0)	n.s
Skeletal maturity, *n*%	35.3%	35.3%	n.s
Side of knee, *n*			0.569 ^a^
Left	9	15	
Right	8	19	
Interval time, month	46 (IQR, 18)	-	-

IQR, interquartile range; interval time, the time from the first dislocation to the contralateral dislocation; ^a^, side of the knee joints at the first patellofemoral dislocation, showing the results of the Chi-square test; n.s, not statistically significant.

**Table 2 jcm-12-00180-t002:** Inter-and intra-observer reliability of each parameter, showing the ICC and 95% confidence interval.

	Inter-Observer 1	Inter-Observer 2	Intra-Observer 1	Intra-Observer 2
Dejour classification ^a^	0.91 (0.85, 0.95)	0.94 (0.88, 0.97)	0.93 (0.85, 0.96)	0.87 (0.74, 0.94)
LTI	0.88 (0.83, 0.93)	0.85 (0.77, 0.92)	0.86 (0.79, 0.91)	0.90 (0.85, 0.94)
Trochlear depth	0.78 (0.66, 0.85)	0.80 (0.68, 0.89)	0.82 (0.77, 0.90)	0.80 (0.69, 0.88)
TT-RA distance	0.91 (0.83, 0.93)	0.88 (0.84, 0.92)	0.83 (0.75, 0.91)	0.95 (0.92, 0.98)
Total femoral torsion	0.93 (0.88, 0.96)	0.94 (0.89, 0.97)	0.85 (0.74, 0.92)	0.93 (0.89, 0.97)
Neck torsion	0.95 (0.92, 0.98)	0.94 (0.91, 0.96)	0.96 (0.94, 0.98)	0.97 (0.95, 0.98)
Mid torsion	0.93 (0.88, 0.97)	0.91 (0.85, 0.94)	0.90 (0.82, 0.94)	0.92 (0.87, 0.98)
Distal torsion	0.95 (0.91, 0.98)	0.94 (0.90, 0.97)	0.90 (0.83, 0.94)	0.96 (0.94, 0.98)
Knee joint rotation	0.93 (0.88, 0.97)	0.89 (0.79, 0.94)	0.93 (0.85, 0.97)	0.96 (0.92, 0.99)
Tibial torsion	0.96 (0.94, 0.99)	0.93 (0.90, 0.96)	0.97 (0.95, 0.99)	0.92 (0.86, 0.95)
PCA	0.94 (0.89, 0.96)	0.93 (0.85, 0.97)	0.87 (0.78, 0.92)	0.96 (0.93, 0.99)
TEW	0.96 (0.93, 0.99)	0.92 (0.83, 0.96)	0.91 (0.80, 0.95)	0.94 (0.87, 0.99)
Patellar height	0.90 (0.83, 0.96)	0.89 (0.81, 0.94)	0.82 (0.74, 0.91)	0.91 (0.82, 0.98)
HKA angle	0.95 (0.87, 0.98)	0.92 (0.83, 0.97)	0.87 (0.75, 0.94)	0.92 (0.88, 0.97)
Patellar tilt	0.88 (0.81, 0.93)	0.86 (0.77, 0.94)	0.91 (0.84, 0.95)	0.93 (0.84, 0.97)
Congruence angle	0.87 (0.81, 0.95)	0.91 (0.85, 0.96)	0.92 (0.83, 0.96)	0.89 (0.82, 0.94)
Patellar displacement	0.93 (0.85, 0.97)	0.91 (0.85, 0.94)	0.88 (0.78, 0.93)	0.92 (0.86, 0.97)

LTI, lateral trochlea inclination; TT-RA, tibial tubercle to Roman arch distance; PCA, posterior condylar angle; TEW, transepicondylar width; HKA, hip-knee-ankle angle; ICC, intra-class correlation coefficient, with a value of >0.75 indicating excellent agreement; ^a^, results of weighted kappa analysis.

**Table 3 jcm-12-00180-t003:** Differences in the parameters between the study group and the control group via the Wilcoxon rank-sum test, showing the median and IQR.

	Study Group	Control Group	*p* Value
LTI, °	11.9 (9.4–13.6)	13.9 (11.1–16.2)	**<0.001**
Trochlear depth, mm	3.8 (2.6–4.2)	4.1 (3.3–4.5)	**<0.001**
TT-RA distance, mm	24.1 (20.9–25.7)	19.5 (17.9–22.2)	**<0.001**
Total femoral torsion, °	19.2 (14.7–22.7)	18.5(16.6–21.3)	0.551
Neck torsion, °	23.3 (20.2–25.3)	22.3 (19.1–25.4)	0.375
Mid torsion, °	27.1 (26.3–29.5)	27.2 (24.6–29.8)	0.302
Distal torsion, °	14.4 (11.1–16.6)	14.6 (10.9–18.3)	0.057
Knee joint rotation, °	9.5 (7.3–12.1)	9.4 (7.4–12.6)	0.295
Tibial torsion, °	32.1 (28.7–35.5)	30.8 (23.1–32.7)	**0.001**
PCA, °	4.4 (2.2–5.4)	4.4 (2.8–4.7)	0.867
TEW, mm	72.7 (70.2–74.6)	73.1 (70.3–74.9)	0.122
Patellar height	1.32 (1.22–1.39)	1.34 (1.12–1.50)	0.863
HKA angle, °	1.9 (1.3–2.8)	0.8 (−0.5–1.8)	**<0.001**
Patellar tilt, °	23.9 (20.9–33.1)	17.8 (13.5–22.4)	**<0.001**
Congruence angle, °	10.2 (9.0–13.0)	9.2 (6.5–10.5)	**<0.001**
Patellar displacement, mm	9.9 (9.3–11.0)	7.6 (6.4–9.9)	**<0.001**
TT-RA/TEW, %	32.7 (29.8–34.9)	27.6 (24.6–31.3)	**<0.001**
Dejour classification (A/B/C/D)	1/2/7/7	4/12/12/6	**<0.001 ^a^**

LTI, lateral trochlea inclination; TT-RA, tibial tubercle to Roman arch distance; PCA, posterior condylar angle; TEW, transepicondylar width; HKA, hip–knee–ankle angle; IQR, interquartile range; ^a^, results of the Chi-square test. Bold indicates statistical significance at *p* < 0.05.

**Table 4 jcm-12-00180-t004:** Diagnostic ability of the parameters for subsequent contralateral LPD.

Variables	AUC	*p* Value	Cutoff Value	Sensitivity, %	Specificity, %
LTI	0.669	0.051	-	-	-
Trochlear depth	0.593	0.285	-	-	-
TT-RA distance	0.727	0.009	20.0	82.3	44.1
Distal torsion	0.548	0.583	-	-	-
Tibial rotation	0.545	0.603	-	-	-
HKA angle	0.797	<0.001	1.3	82.4	70.6
Patellar tilt	0.703	0.019	19.0	88.2	50.0
Congruence angle	0.725	0.009	8.7	82.4	51.1
Patellar displacement	0.817	<0.001	9.2	88.2	64.7
TT-RA/TEW	0.741	0.006	29.5	83.3	61.7

LPD, lateral patellar dislocation; LTI, lateral trochlea inclination; TT-RA, tibial tubercle-Roman arch distance; HKA, hip–knee–ankle angle; TEW, transepicondylar width; AUC, area under the curve, with a value more than 0.7 indicating fair to good diagnostic capacity.

## Data Availability

The data associated with the paper are not publicly available but are available from the corresponding author upon reasonable request.

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
