# Peer review of "Prediction of Subsequent Contralateral Patellar Dislocation after First-Time Dislocation Based on Patellofemoral Morphologies"

_jcm, 2022, doi:10.3390/jcm12010180_

Round 1
Reviewer 1 Report
This seems to be a self citing measure without reference to traditional patellofemoral measures or historical citations.
If you want to report your own measurements, the wider scientific community would prefer traditional patellofemoral indices to be included in your analysis
Reviewer 2 Report
Radiographic - X ray and CT - imaging and analysis - is detailed and appropriate.
However two issues.
Firstly , I am not sure that the findings will influence prophylactic management of the contralateral side.
The second issue is that I am concerned about CT imaging in these young patients- this is not without risk . I presume informed consent for this study was waived by the institutional review board because it had been obtained at the initial presentation ?
Reviewer 3 Report
The paper is well written and structured but from my point of view the novelty and the informations provided are not very useful in practice. You presented a study that shows the causes of lateral dislocation of patella which are well known and the incidence of contralateral dislocation due to anatomy abnormalities. I do not see the point of this study in practice. It is important just for statistics and to inform the patient about the risk of patella dislocation.
In conclusion you have a good technical paper but with poor interest to the readers.
Reviewer 4 Report
Thank you for the pleasure to review your interesting article "Prediction of Subsequent Contralateral Patellar Dislocation After First-Time Dislocation Based on Patellofemoral Morphologies." It is a study investigating 17 patients with contralateral LPD against 34 control patients and measured radiographic parameters after 1st ipsilateral LPD to predict contralateral LPD.
Overall - The English requires improvement. I recommend using a native English speaker to correct the phrasing.
Introduction
Line 43 - is there a reference for this impact on the patient?
Line 69-72 - I am confused as to what the aim of this study is. Is it to investigate factors for contralateral LPD? or just LPD in general? Given the ambiguity, the structure of the introduction is confusing. The paragraphs on their own flow well with a clear point made but when put together are confusing. The first paragraph starts with contralateral LPD incidence, then LPD risk factors, then contralateral LPD risk factors again and then purposes. If this is a paper investigating contralateral LPD as suggested in the abstract and title, I question whether the 2nd paragraph is necessary for the introduction. Additionally line 45 would be best suited for the 3rd paragraph
Methods
Overall well explained and measurements clearly outlined. Great images to help understand and replicate the same measurements for others. One concern is, I understand the TT-RA is a novel measurement proposed by your institution. I could not find its widespread use outside of your institution in the literature and worry about the bias that may be involved with this measurement.
Line 94 - what is the follow up of the 17 patients included and the 34 control patients?
line 95 - what is your follow up protocol?
Line 97-103 - are bilateral full lower limb CTs routine? How was the process of getting this approved by your ethics board? Has there been a cost benefit analysis for its justification?
Again - is it routine to due full leg length XR on all patients after isolated dislocations? This is expensive and requires specific radiographic machines. How was this justified by your hospital and ethics if only 5% have previously shown risk for contralateral LPD?
Line 192 - it would of been ideal if there was at least 3 observers. were there any circumstances where there was large discordance amongst measurements?
Results
line 211- what was the follow up time for the control group
Table 2 and line 218 - can you please explain how >0.75 is excellent agreement? With this definition every measurement was excellent?
Discussion
Line 282-283 - I am unsure that words such as "responsible" can be used. These are risk factors that are associated with contralateral LPD not the causes. Please be wary of such strong language.
Line 293- do you mean "recognition" not cognition?
Line 297- why do you think that patella height was not a risk factor in this study? please discuss --> it appears to be discussed in line 349-357. Great work! However, confused with the statement in line 350-352 as it seems to contradict your statement in line 297 on your previous paper (reference 11). Why do you believe your previous paper showed relevance and this paper doesnt? You still have not answered/explained this question and rather only stated that the BlackBurne-Peel index can be comparable.
Line 320-321- though previously elaborated in a seperate paper, it would be helpful for the readers if you outlined why TT-TG is not correlated to the contralateral side and TT-RA is.
conclusion
line 375 - was not >0 degrees defined as valgus? Didnt the control group have 0.8 degrees median? this should read "increased valgus ..."
Round 2
Reviewer 3 Report
It is a well written article but with low scientific soundness.